# The Blob marine heatwave transforms California kelp forest ecosystems

Kristen M. Michaud [1✉], Daniel C. Reed[1] & Robert J. Miller[1]

Ocean warming has both direct physiological and indirect ecological consequences for marine organisms. Sessile animals may be particularly vulnerable to anomalous warming given constraints in food acquisition and reproduction imposed by sessility. In temperate reef ecosystems, sessile suspension feeding invertebrates provide food for an array of mobile species and act as a critical trophic link between the plankton and the benthos. Using 14 years of seasonal benthic community data across five coastal reefs, we evaluated how communities of sessile invertebrates in southern California kelp forests responded to the "Blob", a period of anomalously high temperatures and low phytoplankton production. We show that this event had prolonged consequences for kelp forest ecosystems. Changes to community structure, including species invasions, have persisted six years post-Blob, suggesting that a climate-driven shift in California kelp forests is underway.

[1] Marine Science Institute, University of California Santa Barbara, Santa Barbara, CA 93106, USA.  ✉email: kmmichaud@ucsb.edu

As oceans warm, marine communities are exposed to novel and stressful abiotic and biotic conditions. Such is the case during acute periods of anomalous ocean warming—marine heatwaves—which are predicted to increase in severity and frequency over the next century due to anthropogenic climate change[1,2]. Recent marine heatwaves have caused extensive ecological and socioeconomic damage to communities in tropical and temperate ecosystems worldwide[3], including severe coral bleaching[4] and massive die offs of seagrass and kelp[5,6]. Marine heatwaves directly stress organisms via the lethal and sublethal physiological consequences of increased temperatures[4,7], but also have more indirect ecological consequences. For example, prolonged warming and resultant ocean stratification can retard upwelling of deep, nutrient-rich water to sunlit surface waters[8,9]. Temperate coastal ecosystems rely on upwelled nitrogen to fuel primary production by phytoplankton that sustain food webs[10,11] and macrophytes such as seagrasses and kelps that provide critical habitat for diverse marine communities[12,13]. Consequently, the extremely productive coastal ecosystems in upwelling regions may be particularly susceptible to ecological damage by marine heatwaves.

Sessile invertebrates provide food for an array of mobile invertebrate and vertebrate predators on shallow temperate rocky reefs, acting as a critical trophic link between the plankton and the benthos through suspension feeding[10,14]. Their dependence on plankton may make this diverse group of primary space holders particularly vulnerable to the effects of marine heatwaves and continued ocean warming[15], and a recent meta-analysis found that sessile species are generally more susceptible to the adverse effects of heatwaves than mobile species[16]. Most of the research done to date on the effects of marine heatwaves on sessile invertebrates has focused on hermatypic corals in the tropics, while on temperate reefs many studies have emphasized habitat-forming macroalgae and seagrasses rather than sessile invertebrates[4–6,15–17]. Given their ecological importance in temperate systems and vulnerability to environmental change, a more comprehensive assessment of the effects of anomalous warming on these diverse communities seems warranted.

In 2014–2015, a persistent high-pressure zone pinned warm water masses along North America's west coast; the resulting extreme heatwave was unprecedented in recorded history and became known as the "Blob"[9,18,19]. Positive temperature anomalies were accompanied by negative chlorophyll-a anomalies across the Southern California Bight in the winter and spring of 2014 and 2015[19], due to deepening of the thermocline and nitracline and increased stratification that limited nutrient enrichment of the photic zone[9]. Here we use 14 years of seasonal data to evaluate the resilience of communities of sessile suspension-feeding invertebrates on reefs in the Santa Barbara Channel to the anomalous ocean conditions associated with the Blob. Sessile invertebrate abundance and species richness declined significantly, and species composition changed during the heatwave in response to depressed phytoplankton abundance that compounded the detrimental effects of warming. These changes, including greater abundance of nonindigenous and warmer-water species, have persisted 6 years post-Blob, suggesting that a prolonged climate-driven shift in the community structure of California kelp forests is underway.

## Results and discussion

The Santa Barbara Coastal Long Term Ecological Research program has monitored benthic communities in five kelp forests seasonally since 2008 using fixed transect diver surveys, and moored sensors at each reef have recorded bottom temperatures every 15 min. Blob-associated positive bottom temperature anomalies began in winter 2014 and persisted through autumn 2016 (Fig. 1a)[18]. Peak temperature anomalies occurred during the summer and autumn of 2014 and 2015 (Fig. 1a), and the average temperature anomaly in autumn 2015 was +3.1 °C, equivalent to an average daily temperature of 19.6 °C. In 2014 and 2015, 91 and 69% of autumn days, respectively, were classified as heatwave days as defined by Hobday et al.[20]. Seasonal chlorophyll-a concentration, a proxy for phytoplankton abundance, was obtained from satellite imagery at each of the five reefs over the 14-year period. The average chlorophyll-a concentration was anomalously low throughout the warming period, and exceptionally low during the springs of 2014 and 2015 (Fig. 1a), when upwelling-driven nutrient enrichment typically supports dense phytoplankton blooms.

Mean sessile invertebrate cover averaged across all sites declined 71% during the Blob, reaching a 14-year minimum of 7% in autumn of 2015 (Fig. 1b and Supplementary Fig. 1). Species richness declined 69% during the same period (Fig. 1b and Supplementary Fig. 1). The responses of invertebrates to warming were not consistent across time even though the duration and intensity of warming was similar in 2014 and 2015, suggesting that extended periods of elevated seawater temperature were not solely responsible for the most severe loss of invertebrates. For ectotherms, increases in ambient seawater temperature should be met with increases in metabolic rate and food requirements to sustain metabolism[21]. Because of their sedentary lifestyle, sessile invertebrates cannot actively forage for food or seek spatial refuge from thermal extremes, and limitations in their planktonic food supply can result in metabolic stress over extended periods[22,23]. Anomalously low chlorophyll-a concentrations during the Blob (Fig. 1a), particularly in the spring of 2015, indicated that food limitation was a likely driver of invertebrate decline. Results from piecewise structural equation modeling (Fig. 2) that incorporated biological interactions with space competitors (understory macroalgae), predators (sea urchins), and foundation species (giant kelp) showed that the severity of warming had both a direct and indirect effect on the sessile invertebrate community. The proportion of heatwave days was a direct negative predictor of sessile invertebrate cover (−0.11) and species richness (−0.21). The proportion of heatwave days was an even stronger negative predictor of chlorophyll-a concentration (−0.26), yielding negative indirect effects on invertebrate cover (−0.07) and species richness (−0.05) due to the positive influence of chlorophyll-a concentration on sessile invertebrate cover (+0.26) and richness (+0.20).

Consequences of heatwaves on benthic community structure can be difficult to predict if community-level interactions exacerbate or ameliorate stressful conditions[24,25]. The biomass of giant kelp, Macrocystis pyrifera, an important foundation species on temperate reefs, was not strongly impacted by the Blob in the Santa Barbara Channel (Fig. 2)[18,26], but experienced warming-induced declines at its equatorward range edge[26,27]. Given the strong positive relationship between giant kelp biomass and sessile invertebrate cover (+0.40) and species richness (+0.46), future declines in giant kelp during warming events, as has been seen in other kelp forest systems[5,17], will likely contribute to sessile invertebrate declines. Similarly, loss of kelp canopy has been shown to increase understory algal cover and biomass[28,29], a strong negative predictor of invertebrate cover (−0.45) and species richness (−0.32), apparently due to increased competition between understory algae and invertebrates for space.

Invertebrate phyla did not respond equally to the Blob; some phyla experienced severe declines during the warming event while others resisted change. Bryozoans, sponges, annelids, and ascidians were particularly hard-hit and declined to their lowest percent cover during the Blob (Fig. 3, Supplementary Fig. 2). For

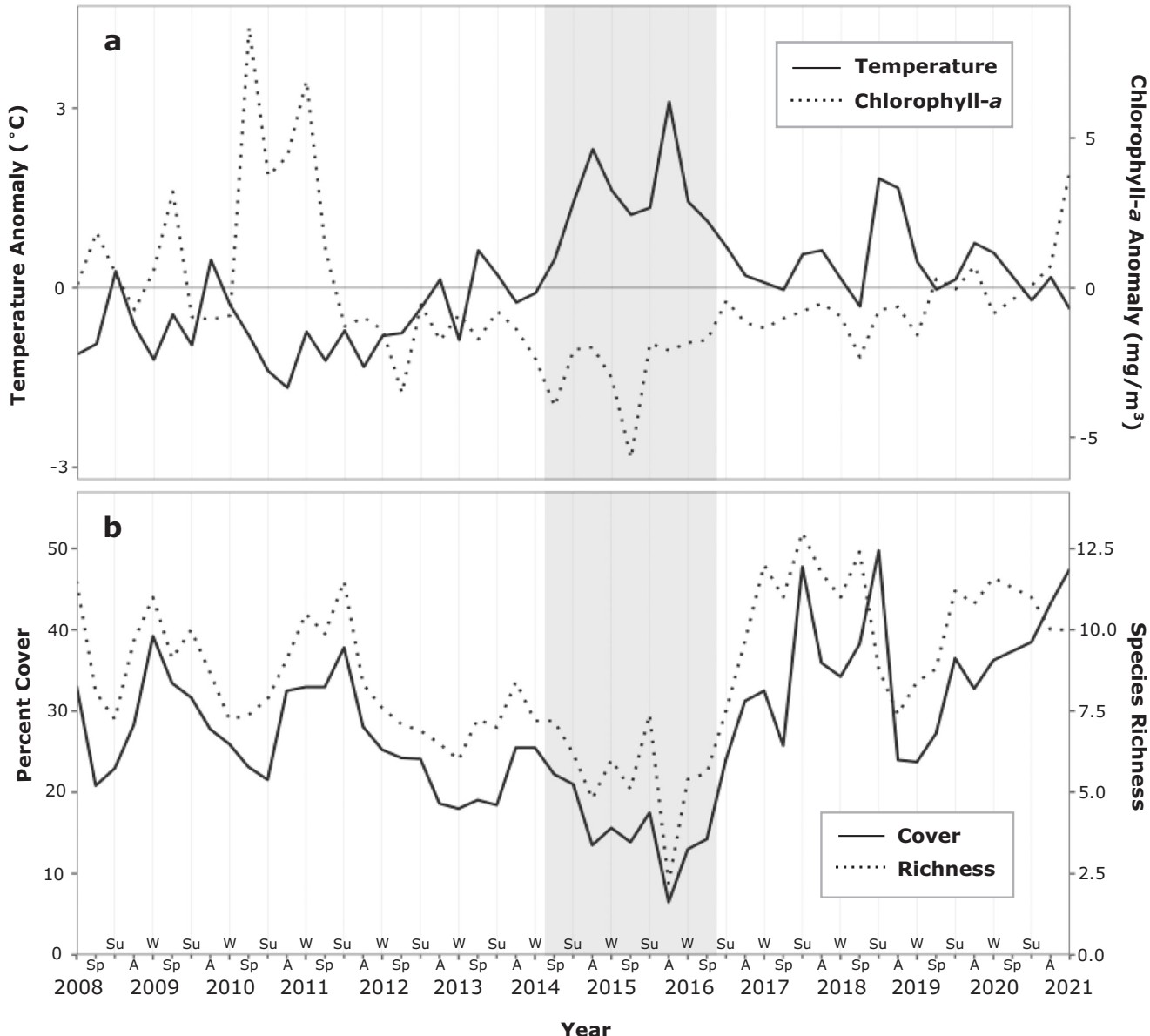

**Fig. 1 Average seasonal bottom temperature anomaly, chlorophyll-a concentration anomaly, and percent cover and species richness of sessile invertebrates across five sites.** The Blob, an anomalous warming period from spring of 2014 to winter of 2016, is highlighted in gray, coincident with (**a**) positive temperature anomalies (°C; solid line), negative chlorophyll-*a* anomalies (mg/m³; dashed line), and declines in (**b**) invertebrate cover (solid line) and species richness (number of unique species/taxa/80 contact points; dashed line). Seasons are denoted by Sp (Spring), Su (Summer), A (Autumn) and W (Winter).

example, in autumn 2015 bryozoans and sponges were not recorded at any site while ascidians were recorded at only one site in very low abundance (i.e. ~1% cover). The decline in the percent cover of annelids began prior to the Blob, making it difficult to attribute their low abundance in 2015 solely to warming. Bryozoans and annelids rapidly recovered following the Blob, and by 2017 their percent cover approached or surpassed pre-heatwave levels. By contrast, the recovery of sponges was considerably slower, with pre-Blob levels not evident until 2020 (Fig. 3, Supplementary Fig. 2). Mollusks were relatively unaffected by the heatwave, although their highest cover was observed in the years following the Blob, suggesting that they may have benefited from delayed indirect positive effects of warming (Fig. 3, Supplementary Fig. 2, Supplementary Fig. 4).

Although the percent cover of most sessile invertebrate phyla returned to pre-Blob levels within a couple of years after the

heatwave, changes in the species composition of the sessile invertebrate community attributed to the Blob have persisted (PERMANOVA, $F = 13.462$, $p < 0.001$; Fig. 4). As positive temperature anomalies subsided in early 2016, invertebrate cover and richness steadily increased over time, peaking in the summers of 2017 and 2018 (Fig. 1). This increase in invertebrate cover was largely driven by increases in the abundance of bryozoans, and to a lesser extent, mollusks (Fig. 3, Supplementary Fig. 2, Supplementary Fig. 3, Supplementary Fig. 4). Two invasive bryozoan species accounted for much of this increase (Fig. 4). The percent cover of *Watersipora subatra*, a recent invader in the Santa Barbara Channel[30], increased after the Blob (Supplementary Fig. 3, Fig. 4; IndVal = 0.455, $p < 0.001$), while *Bugula neritina*, a long-established invader, is now substantially more abundant than its morphologically similar native relative, *Bugulina californica* (Supplementary Fig. 3, Fig. 4; IndVal = 0.549, $p < 0.01$).

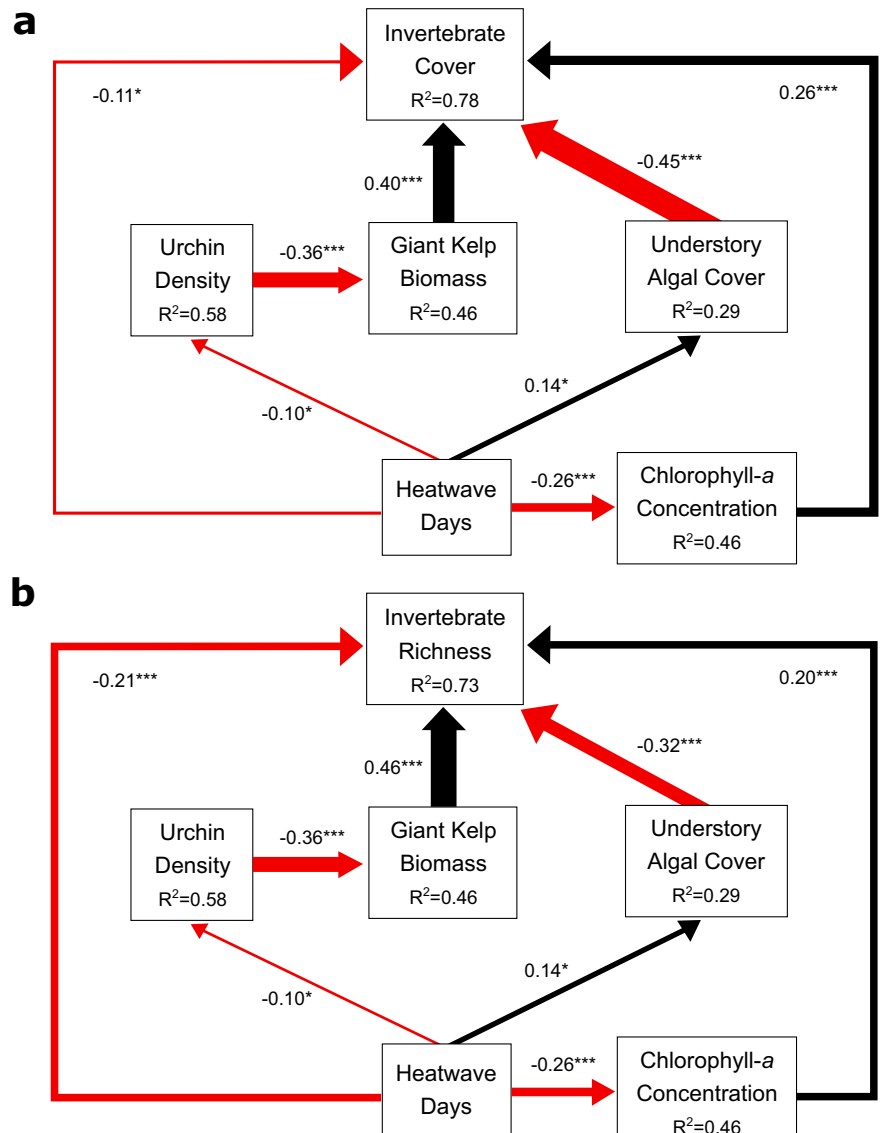

**Fig. 2 Piecewise structural equation modeling (SEM) for sessile invertebrate cover and species richness.** Arrows indicate directionality of effects on (**a**) invertebrate cover and (**b**) species richness. Red arrows show negative relationships; black arrows show positive relationships. $R^2$ values are conditional $R^2$. Arrow widths are proportional to effect sizes as measured by standardized regression coefficients (shown next to arrows). ***$p < 0.001$, **$p < 0.01$, *$p < 0.05$. Insignificant pathways are not included.

The increased abundance of *W. subatra* and *B. neritina* following the Blob could be due to reductions of specialized native species that otherwise would outcompete generalist invaders[31], increased tolerance to thermally or metabolically stressful conditions, as has been demonstrated for *W. subtorquata* and some invasive ascidian species[32], or increased recruitment of these species[33].

In addition to increases in nonnative species, the abundance of a native southern-affinity sessile gastropod, *Thylacodes squamigerus*, increased significantly since the onset of the Blob (IndVal = 0.711, $p < 0.001$; Fig. 4, Supplementary Fig. 4). As one of the few locally abundant invertebrates with a southern range extending beyond Baja California, *T. squamigerus* may have been preadapted to warmer temperatures. *T. squamigerus* also has the capacity to consume kelp detritus as an alternative food source[34], which may have facilitated its survival during extended periods of low plankton availability.

Though most sessile invertebrates are suspension feeders relying on the delivery of plankton and particulate organic matter for food[10,11], there are differences in life history traits and feeding strategies among benthic phyla that may result in unequal responses to marine heatwaves. Colonial species, including sponges, most bryozoans, and many ascidians, are generally shorter-lived and exhibit rapid growth rates. By contrast, anthozoan cnidarians and mollusks are generally longer-lived with slower growth[35], and may have specific traits (e.g., lower metabolism, energy stores, alternate feeding strategies) that enable them to survive prolonged periods of warming with anomalously low phytoplankton supply. For example, anemones are opportunistic feeders that can consume zooplankton and detritus[36], while colonial species such as bryozoans and ascidians may be more dependent on smaller phytoplankton[37], with a lower capacity to switch food sources in the event of low phytoplankton abundance. Though adult anthozoans and mollusks were generally resilient during the Blob, prolonged reductions in phytoplankton could adversely affect their populations by limiting recruitment in species with planktotrophic larvae that depend on phytoplankton for food. Increases in bryozoan cover following the Blob, particularly in two invasive species, suggests that rapid

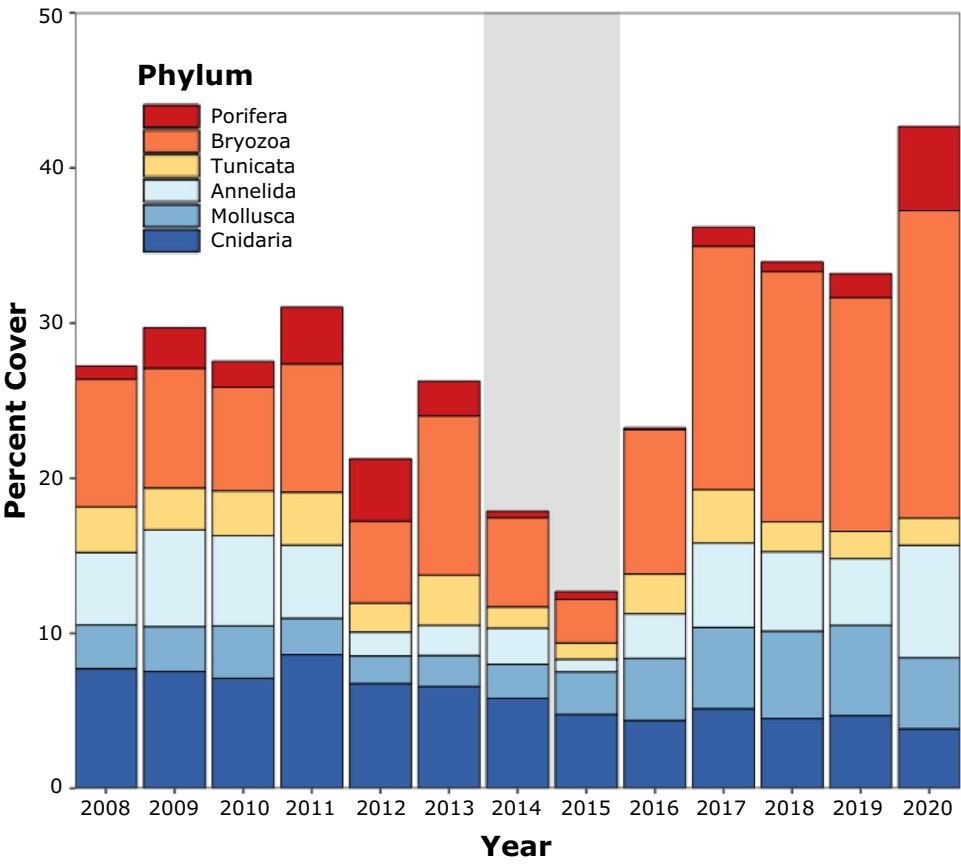

**Fig. 3 Average annual percent cover of invertebrate phyla.** Gray shading indicates the Blob period of 2014– 2015 when bryozoan, ascidian, sponge (Porifera) and annelid cover declined dramatically.

reproduction and growth may have facilitated their colonization following disturbance and increases in unoccupied space. Similar trends were observed in Mediterranean temperate reefs where traits associated with rapid growth and reproduction were significantly more common in benthic assemblages following heatwaves[38].

Suspension-feeding invertebrates are an essential link between pelagic and benthic food webs[10], are important fisheries and aquaculture species, and are especially susceptible to environmental stressors due to the constraints in food acquisition and reproduction imposed by sessility. We found large negative impacts of warming and accompanying phytoplankton decreases on benthic suspension feeders along with persistent changes in their species composition in kelp forests of southern California, even though the major habitat-forming species, giant kelp, was relatively resilient to warming[18,26]. The Blob had similar effects in altering the species composition (but not community biomass) of understory macroalgae at our study sites[18,39], which in turn, may have indirectly affected the sessile invertebrate community. We predict that increasing marine heatwaves will result in future losses and changes in species assemblages of sessile animals on temperate rocky reefs worldwide, and these transformations will likely be exacerbated by warming-induced declines in structure-forming kelps and understory macroalgae. Declines or changes in the species composition of suspension feeders may disrupt coastal food webs if the "winners" are functionally dissimilar from existing taxa. For example, heatwaves have altered functional traits in temperate reef systems through changes to benthic community structure[38]. Additionally, introductions of tropical herbivorous fish to temperate reefs during reef tropicalization have resulted in herbivore-mediated declines of macroalgae[40],

altering ecosystem function. Such functional shifts may have significant consequences for temperate reefs, including decoupling of primary production from higher trophic levels that depend on these vital primary consumers for survival.

## Methods

**Study system.** The five subtidal reef sites were located along the mainland of the Santa Barbara Channel within the northern portion of the Southern California Bight: Carpinteria (119.54 °W, 34.39 °N), Mohawk (119.73 °W, 34.39 °N), Isla Vista (119.86 °W, 34.40 °N), Naples (119.95 °W, 34.42 °N), and Arroyo Quemado (120.12 °W, 34.47 °N). Sites ranged in depth from 5.4 to 7.5 m on rocky substrate (shale, sandstone, mudstone). All sites and transects were monitored seasonally by the Santa Barbara Coastal (SBC) Long Term Ecological Research (LTER) program.

**Oceanographic anomalies.** Bottom temperature at each site was recorded using loggers fixed to the bottom (Stowaway Onset tidbits, Onset Computer, Bourne, Massachusetts, USA). Temperature measurements recorded once every 15 min were used to calculate the mean temperature for each season from 2003 through winter of 2021[41]. Seasonal temperature anomalies for each site were calculated as the mean of a given season in a given year minus the mean of that season averaged over all years from 2003 through 2021. Seasonal temperature anomalies for the study region were calculated as the mean seasonal anomalies averaged across the five study sites.

Heatwaves were identified as 5 or more consecutive days with daily bottom temperatures above the 90th percentile[20] for the period 2003–2021. The number of heatwave days at each site were summed within each season in the time series and used to calculate the proportion of heatwave days per season at each site.

To calculate the seasonal chlorophyll-*a* concentration at each site, the average monthly chlorophyll-*a* concentration was extracted from the mapped chlorophyll data from the MODIS (Moderate Resolution Imaging Spectroradiometer) instrument aboard the Aqua satellite[42]. Chlorophyll-*a* values within 4km-square bins were extracted at each site centered 2 km south (offshore) of the reef's location to prevent bin overlap with land. Seasonal chlorophyll-*a* values were calculated using the average chlorophyll-*a* value at each reef over 3 months (winter = December, January and February; spring = March, April and May; summer = June, July and August; autumn = September, October and November).

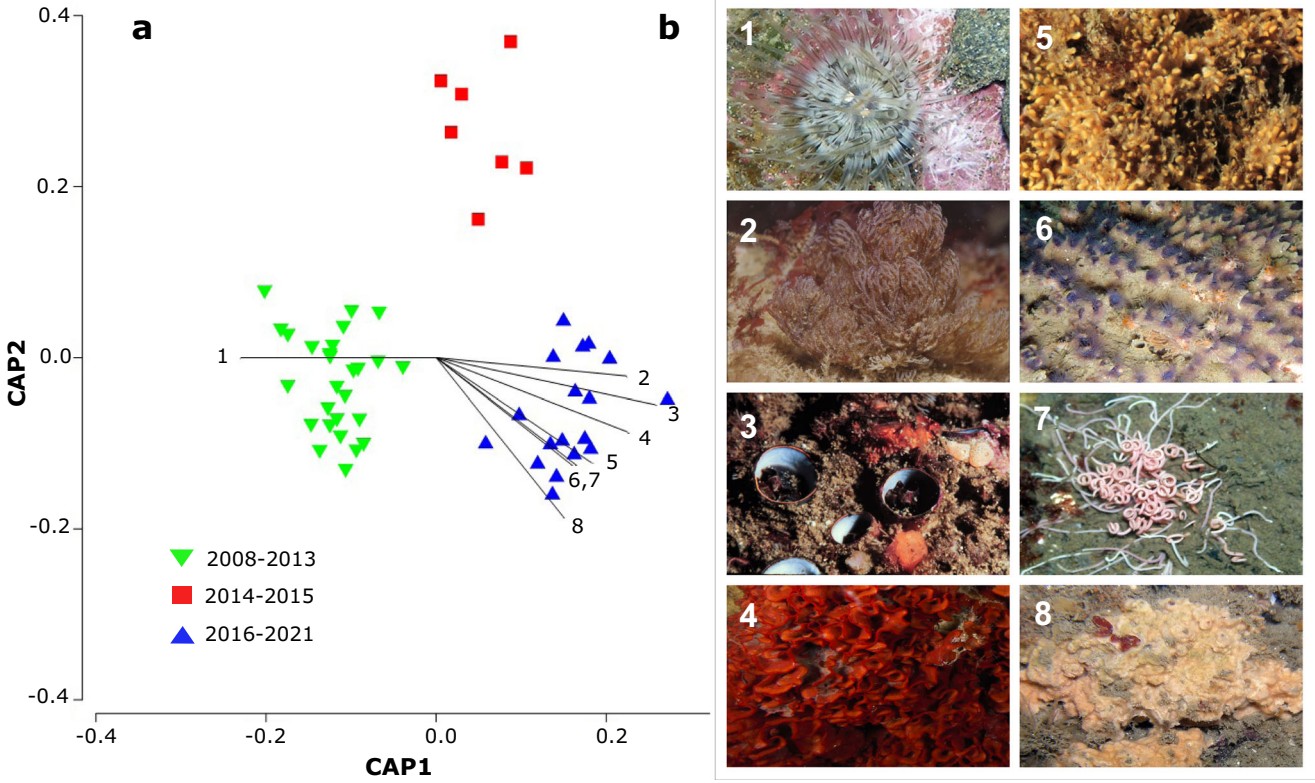

**Fig. 4 Canonical Analysis of Principal Coordinates (CAP) of species composition of kelp forest sessile invertebrate communities. a** CAP with time relative to the Blob (before: 2008–2013, during: 2014–2015, and after: 2016–2021) as a constraining variable. Points represent the average species composition across four sites for each sampling season. Lines represent (**b**) prominent species with correlations >0.6; (1) *Actinostella californica* (Phylum Cnidaria) (2) *Bugula neritina* (Phylum Bryozoa) (3) *Thylacodes squamigerus* (Phylum Mollusca) (4) *Watersipora subatra* (Phylum Bryozoa) (5) *Diaperoforma californica* (Phylum Bryozoa) (6) *Phragmatopoma californica* (Phylum Annelida) (7) *Timarete luxuriosa* (Phylum Annelida) (8) Encrusting bryozoan spp. (Phylum Bryozoa).

**Benthic community sampling**. Sessile invertebrate and understory macroalgal communities were sampled twice seasonally from 2008 to 2013 and once seasonally from 2013 to winter of 2021[43]. Data were not collected during the spring of 2020 due to research restrictions stemming from the COVID-19 pandemic. Uniform point contact surveys were performed by divers along a 40 m x 2 m permanent transect. Paired points were sampled every 0.5 m for a total of 80 points per transect. Sessile invertebrates were identified to species or family level. All species intersecting the point were recorded, such that the total percent cover of all species combined could exceed 100%, while the cover of any individual species could not exceed 100%. A complete list of invertebrate species sampled can be found in Supplementary Table 1. The total number of unique species encountered in 80 points per transect was used to measure sessile invertebrate species richness. Giant kelp (*Macrocystis pyrifera*) was sampled along the same permanent transect and the total number of plants and stipes per plant were recorded and converted to biomass using established allometric relationships[44]. The density of red (*Mesocentrotus franciscanus*) and purple (*Strongylocentrotus purpuratus*) urchins was sampled in six permanent 1 m$^2$ quadrats per transect, and the average density across quadrats was calculated for each transect[45].

**Statistics and reproducibility**. All piecewise structural equation models (SEM) analyses were performed in R v 4.1.2[46]. Piecewise structural equation models were evaluated using the **piecewiseSEM** package[47]. To evaluate how the percent cover (Fig. 2a) and species richness (Fig. 2b) of sessile animals were influenced by the severity of warming (measured as the proportion of heatwave days per season) and by food availability (chlorophyll-*a* concentration, mg/m$^3$) we used piecewise structural equation modeling. Established predictors of invertebrate cover such as biomass of the foundation species, giant kelp (*Macrocystis pyrifera*) (g dry mass/m$^2$), urchin density (red and purple urchins; average number of individuals/m$^2$), and understory macroalgal cover[12,48] were included in the models. Two separate models were developed to evaluate the environmental effects on invertebrate cover and species richness. Prior to inclusion in the model, proportional cover of invertebrates and understory macroalgae and invertebrate species richness were square root transformed to improve normality. The concentration of chlorophyll-*a* and giant kelp biomass were log-transformed to reduce the effects of extreme values. All community metrics were averaged for each sampling season at each site for years with more than

one sampling event per season. All fixed effects were then centered and scaled by subtracting the factor mean from each factor and dividing by the factor standard deviation.

Models evaluated the effects of environmental and community-level variables on: (1) sessile invertebrate cover (Fig. 2a), and (2) species richness (Fig. 2b). Site was included as a random effect in each model, while the proportion of heatwave days per season, the average seasonal chlorophyll-*a* concentration, giant kelp biomass, understory macroalgal percent cover, red and purple urchin density, and season were included as fixed effects. Temporal autocorrelation did not improve the Akaike Information Criterion (AIC) for the models and was excluded. Hypothesized pathways can be found in Supplementary Table 2 and Supplementary Table 3.

The effect of the anomalous warming period from 2014–2015 on community composition was evaluated using Canonical Analysis of Principal Coordinates (CAP) as described by Anderson and Willis[49] in **PRIMER 7 PERMANOVA +**[50]. Sampling events were partitioned into time periods: before the anomalous warming period (Before, winter 2008–winter 2014), during the anomalous warming period (During, spring 2014–winter 2016), and after the warming period (After, spring 2016–winter 2021) and time period was treated as a constraining variable. Samples from the Isla Vista site were excluded in this analysis, since sampling did not include the years 2008–2011. The average community composition across the remaining four sites were used for the analysis. The community matrix was then square root transformed and Bray-Curtis dissimilarities were used as a resemblance matrix prior to ordination by CAP (Fig. 4).

To evaluate which species contributed the most to community dissimilarity over time, an Indicator Species Analysis was conducted in R v 4.1.2[46] in the **indicspecies** package[51] using the function *multipatt*. Sites were restricted to the time period (before, during, and after).

Permutational Multivariate Analysis of Variance Using Distance Matrices (PERMANOVA) analyses were performed in R v 4.1.2[46] in the **vegan** package[52]. Differences in species composition were evaluated using Bray-Curtis dissimilarities in the function *vegdist*. Prior to performing PERMANOVA, we evaluated the multivariate homogeneity of group dispersions using the *betadisper* function. There were significant differences detected in distances to spatial medians between time periods (ANOVA, $F = 4.0241$, $p = 0.024$). Pairwise comparisons showed that significant differences were between the Before and During periods ($p = 0.005$),

while there was no significant difference between Before and After ($p = 0.37$) and During and After ($p = 0.076$). The During time period was then removed to satisfy the dispersion homogeneity assumption, and group dispersions were reassessed between Before and After communities. There was no significant difference between the homogeneity of group dispersions between Before and After (ANOVA, $F = 0.8114$, $p = 0.373$), so results of PERMANOVA between Before and After can be interpreted confidently. PERMANOVA was then performed using the function *adonis2* using time period as a fixed effect.

**Reporting summary**. Further information on research design is available in the Nature Research Reporting Summary linked to this article.

## Data availability

All data used are available publicly from the Santa Barbara Coastal Long Term Ecological Research program and from NASA (AQUA/MODIS Chlorophyll Data): SBC LTER: Reef: Bottom Temperature: Continuous water temperature, ongoing since 2000 ver 26. Environmental Data Initiative. https://doi.org/10.6073/pasta/22ed009da1cf41cbf76490ab2c0c5949 SBC LTER: Reef: Long-term experiment: Kelp removal: Cover of sessile organisms, Uniform Point Contact ver 30. Environmental Data Initiative. https://doi.org/10.6073/pasta/1151c1dcf5110432b6d35f7dc00bb834 SBC LTER: Reef: Long-term experiment: Kelp removal: Invertebrate and algal density ver 19. Environmental Data Initiative. https://doi.org/10.6073/pasta/decb1dcc7b35d2ef401d2dd7d79ea257 SBC LTER: Reef: Long-term experiment: biomass of kelp forest species, ongoing since 2008 ver 8. Environmental Data Initiative. https://doi.org/10.6073/pasta/e30eb31ce1f346255910fe17092f00b1 Moderate-resolution Imaging Spectroradiometer (MODIS) Aqua Chlorophyll Data. Available at: doi:10.5067/AQUA/MODIS/L3B/CHL/2018.

## Code availability

All code is publicly available: Michaud, K. M. kristenmichaud/Chl-File-Code-Michaud-et-al. Chl_file_code_Michaud_et_al. (Version v1). Zenodo. https://doi.org/10.5281/zenodo.7117750 (2022). Michaud, K. M. kristenmichaud/Community-Analysis-Michaud-et-al. Community_Analysis (Version v1). Zenodo. https://doi.org/10.5281/zenodo.7117763 (2022).

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

## Acknowledgements

We thank Clint Nelson, Shannon Harrer, Sarah Sampson, Jacob Ogawa, and Francesca Puerzer for field data collection and Li Kui and Margaret O'Brien for data management. Funding for this work was provided by National Science Foundation grant OCE-9982105, National Science Foundation grant OCE-0620276, National Science Foundation grant OCE-1232779, National Science Foundation grant OCE-1831937, National Aeronautics and Space Administration Biodiversity and Ecological Forecasting Program grant NNX14AR62A, and Bureau of Ocean Energy Management grant M15AC00006.

## Author contributions

K.M.M.: conceptualization, methodology, investigation, formal analysis, visualization, writing—original draft; R.J.M.: conceptualization, methodology, investigation, funding acquisition, writing—review and editing; D.C.R.: methodology, investigation, funding acquisition, writing—review and editing.

## Competing interests
The authors declare no competing interests.
