## [Peer Review File · Communications Biology]

Reviewers' comments:

Reviewer #1 (Remarks to the Author):

The authors present a comprehensive and compelling study on the long-last impacts of the 'the blob' marine heatwave on sessile faunal communities in California. Overall, the paper is very well-written, the data are analysed appropriately and the interpretation is sound. I'm sure it will make an excellent contribution to the literature. I do, however, have several comments and concerns that need to be addressed as part of a substantial revision.

1. I felt the introductory text could have placed the study in a wider context and also highlighted the novelty of the work more effectively. Specifically, sessile invertebrates have been relatively understudied in the context of marine heatwave impacts, compared to corals, macroalgae and plankton for example. In the meta-analysis by Smale et al (2019, Nat Clim Change <https://www.nature.com/articles/s41558-019-0412-1>) they showed that sessile inverts were generally negatively impacted by MHWs but the sample size was relatively low. Perhaps use this to highlight the need to focus on this group. Similarly, recent work (e.g. Smith et al 2021, Science, <https://www.science.org/doi/10.1126/science.abj3593>) show that MHWs have major impacts on human society, yet this work is not cited as support for the need to conduct such studies.
2. Line 54 and elsewhere: I think the values for declines in cover and richness are mean values across the five kelp forest reefs? It would be good to state that and also present these mean values with associated SD/SE so the reader can get a handle on how much spatial variability in conditions and responses was observed.
3. The section running line 54 to 64 only tells half of the story of Fig 1. The declines in cover and richness are well described but the bounce-backs following the MHW are not. The gives a rather biased description of the responses and should be discussed here (I realise you discuss individual species further on, but the big picture patterns need to describe the increases too).
4. It seems odd that the LTER monitoring programme quantifies all components of the kelp forest community (e.g. mobile inverts, understorey algae, kelp) and yet only the responses of a subset of the community are analysed and described here. Whilst I think this is fine, it does need to be justified somewhere in the main text
5. The statement ending 'periods' on line 63 should be supported by a reference.
6. On line 100 a PERMANOVA result is given to describe shifts in community structure through time. I think this value is for the main effect of 'time' but the posthoc pairwise comparisons are not given here or in the supp material (i.e. between before-MHW, during and after). As such it is not clear to the reader how the community shifted through the phases.
7. On line 138 the authors state that giant kelp was resilient to warming yet no data on giant kelp is actually presented in the paper, so the statement is unsupported. I think they should either (i) present a time series of giant kelp biomass for the sites (perhaps add a panel to Fig 1) or (ii) cite papers that show this (i.e. Reed 2016 Nat Comm). My preference would be option one so the reader can get a handle on how much changes in kelp biomass (or lack of) as well as phytoplankton productivity may influence patterns.
8. Line 138 to end. I felt that the final statement and discussion was perhaps a little biased and doesn't fully capture the findings of the study. The authors focus on losses of sessile invertebrates but the data indicate huge gains in some species and, overall, the cover and richness of this community is now comparable of even higher than before the heatwave. If the 'new' species are functionally similar to the 'old' ones then the functioning of this community may very well continue much as before,

thereby not disrupting food webs or decoupling pelagic and benthic systems. Of course, if the 'winners' are functionally different then wide-ranging impacts may ensue. The discussion needs more balance and nuance and shouldn't just focus on declines and losses but rather include the gains and focus more on community reconfigurations and the potential for novel functions and interactions. Perhaps examine and cite Verges et al 2014 Proc Roy Soc here if useful (<https://royalsocietypublishing.org/doi/full/10.1098/rspb.2014.0846>)

Reviewer #2 (Remarks to the Author):

The manuscript by Michaud et al. assesses changes in community assemblages across a 18 year time period, finding that during periods of an extreme marine heatwave in 2014 that community assemblages were altered. I have very few questions and comments, as I have seen one of the authors present a sub-set of this work previously and have disused their methods etc in greater detail with them in person.

The data collected here to make these conclusions is probably second to none in terms of the temporal resolution and length compared to other existing time series in temperate kelp forest ecosystems. This type of data set is what is required in many other locations and I applaud the authors for this long-term approach that would have taken thousands of man hours to complete. For example, in my home country there is perhaps only one data set of 3 or 4 years conducted in a similar way to this. The results are not overly novel, in that such impacts of marine heatwaves have been recorded previously, but the scientific community will gain insights from this work that would justify its publication in this outlet in my opinion.

In saying that, the manuscript is fairly short and it could have benefited from exploration of smaller details. For example, I am slightly biased here in that I was hoping to see greater exploration of trends within different seaweed groups. However, I understand the short format here and I consider the manuscript represents a useful description of the overall trends in community assemblages. The abstract should be altered to indicate the focus on the sessile invertebrates here. A greater focus on the presentation of giant kelp canopy cover could have also been interesting here for the reader.

Specific Comments:

Line 161: 30 year periods are recommended in the Hobbday paper – but I can understand why in situ measurements at 18 years might be better than trying to rely on satellite derived estimates for longer time periods.

Line 189: Spell out SEM first before using.

Figure 3 is interesting, but what about the seaweeds?

All author responses are highlighted in red here and revisions to the manuscript are included in red in the manuscript document.

Reviewer #1 (Remarks to the Author):

1. The authors present a comprehensive and compelling study on the long-last impacts of the ‘the blob’ marine heatwave on sessile faunal communities in California. Overall, the paper is very well-written, the data are analysed appropriately and the interpretation is sound. I’m sure it will make an excellent contribution to the literature. I do, however, have several comments and concerns that need to be addressed as part of a substantial revision.

We thank Reviewer #1 for this feedback, and we attempted to thoroughly address all of their comments and concerns in our revised manuscript.

2. I felt the introductory text could have placed the study in a wider context and also highlighted the novelty of the work more effectively. Specifically, sessile invertebrates have been relatively understudied in the context of marine heatwave impacts, compared to corals, macroalgae and plankton for example. In the meta-analysis by Smale et al (2019, Nat Clim Change <https://www.nature.com/articles/s41558-019-0412-1>) they showed that sessile inverts were generally negatively impacted by MHWs but the sample size was relatively low. Perhaps use this to highlight the need to focus on this group. Similarly, recent work (e.g. Smith et al 2021, Science, <https://www.science.org/doi/10.1126/science.abj3593>) show that MHWs have major impacts on human society, yet this work is not cited as support for the need to conduct such studies.

Excellent suggestion. We added a paragraph in the introduction beginning on Line 33 to provide a broader context for our study and to highlight why sessile invertebrates were specifically investigated. This includes reference to Smale et al. 2019 and their findings that sessile suspension feeding invertebrates may be particularly vulnerable to marine heatwaves and ocean warming. Additionally, we included a reference to socioeconomic impacts of marine heatwaves on Line 23 and cited Smith et al. 2021.

3. Line 54 and elsewhere: I think the values for declines in cover and richness are mean values across the five kelp forest reefs? It would be good to state that and also present these mean values with associated SD/SE so the reader can get a handle on how much spatial variability in conditions and responses was observed.

Yes, these values are overall means across the five reefs. This has been added to the text on Line 71 to clarify this point. Additionally, we have included a supplemental figure (fig. S1) which includes the standard error for the percent cover and richness of invertebrates over time.

4. The section running line 54 to 64 only tells half of the story of Fig 1. The declines in cover and richness are well described but the bounce-backs following the MHW are not. The gives a rather biased description of the responses and should be discussed here (I realise you discuss individual species further on, but the big picture patterns need to describe the increases too).

We originally focused on the declines during the heatwave to focus the story, however, we agree that it is important to discuss the community-wide patterns following the Blob. We have elaborated on this increase in cover across groups in Lines 109-121, as well as discussed potential reasons for it in Lines 151-156.

5. It seems odd that the LTER monitoring programme quantifies all components of the kelp forest community (e.g. mobile inverts, understory algae, kelp) and yet only the responses of a subset of the community are analysed and described here. Whilst I think this is fine, it does need to be justified somewhere in the main text

We have included an additional introductory paragraph to explain why we specifically focused on the responses of sessile invertebrates. Whole community responses to the heatwave were previously assessed by Reed et al. (2016) who examined the response of annual biomass of different functional groups of reef organisms as a function of temperature. Biomass is an important metric for evaluating the trophic position and contribution of an organism or functional group, whereas percent cover provides a more accurate assessment of the primary space occupied by different taxa. At the LTER study sites, the biomass of sessile invertebrates is heavily weighted towards long-lived, large endolithic pholad bivalves, which occupy relatively little primary space on the reef. Thus, while Reed et al. 2016 found that variation in the total biomass of the sessile invertebrate community was unrelated to temperature, we show that the heatwave was a significant predictor of invertebrate percent cover and richness.

Additionally, we included references to Cavanaugh et al. 2019 where the responses of giant kelp to the Blob were assessed across Southern California and Baja, and Lamy et al 2019 who reported on drivers of the macroalgal communities at our study sites, including changes in species composition during the Blob.

6. The statement ending ‘periods’ on line 63 should be supported by a reference.

Thank you for catching this! Two sources (Delorme et al. 2020 and Haider et al. 2020) are now included to support this statement.

7. On line 100 a PERMANOVA result is given to describe shifts in community structure through time. I think this value is for the main effect of ‘time’ but the posthoc pairwise comparisons are not given here or in the supp material (i.e. between before-MHW, during and after). As such it is not clear to the reader how the community shifted through the phases.

We understand the confusion. When determining whether it is appropriate to use PERMANOVA to compare across all three groups, an assumption is that group dispersions are not significantly different. However, when evaluating differences in group dispersions, this assumption was not met when including the “During” time period. Removing “During” results in a PERMANOVA that compares the “Before” community structure to the “After” community structure. Reporting the post-hoc result would be superfluous as the F-statistic indicates that these communities are significantly different from one another. A reference to these methods is made in Lines 267-275, and has been further clarified.

8. On line 138 the authors state that giant kelp was resilient to warming yet no data on giant kelp is actually presented in the paper, so the statement is unsupported. I think they should either (i) present a time series of giant kelp biomass for the sites (perhaps add a panel to Fig 1) or (ii) cite papers that show this (i.e. Reed 2016 Nat Comm). My preference would be option one so the reader can get a handle on how much changes in kelp biomass (or lack of) as well as phytoplankton productivity may influence patterns.

Thank you for catching this. We intended to reference Reed et al. (2016) here, given that we had referenced this study in our earlier statement on the response of *Macrocystis* to heatwaves in local populations. We have included this reference in Line 163. Additionally, we have included reference to Cavanaugh et al. 2019 in Line 163 and in Line 95, which demonstrates that giant kelp was resilient to warming during the Blob in the Santa Barbara Channel. We feel that these papers do much better justice to the giant kelp patterns than we could here.

9. Line 138 to end. I felt that the final statement and discussion was perhaps a little biased and doesn't fully capture the findings of the study. The authors focus on losses of sessile invertebrates but the data indicate huge gains in some species and, overall, the cover and richness of this community is now comparable of even higher than before the heatwave. If the 'new' species are functionally similar to the 'old' ones then the functioning of this community may very well continue much as before, thereby not disrupting food webs or decoupling pelagic and benthic systems. Of course, if the 'winners' are functionally different then wide-ranging impacts may ensue. The discussion needs more balance and nuance and shouldn't just focus on declines and losses but rather include the gains and focus more on community reconfigurations and the potential for novel functions and interactions. Perhaps examine and cite Verges et al 2014 Proc Roy Soc here if useful

(<https://royalsocietypublishing.org/doi/full/10.1098/rspb.2014.0846>)

We expanded the Discussion in Lines 151-156 and 169-174 to address these concerns. Specifically, we discussed how increases in bryozoan cover following the Blob may have been due to functional traits that facilitate rapid growth following disturbance. We also discussed how disruptions to the kelp forest food web will be dependent on functional traits of the new community members, or "winners", as has been seen in other temperate systems. As suggested, we added a reference to Verges et al. 2014 as an example of how tropicalization has disrupted ecosystem function in macroalgal dominated systems.

Reviewer #2 (Remarks to the Author):

10. The manuscript by Michaud et al. assesses changes in community assemblages across a 18 year time period, finding that during periods of an extreme marine heatwave in 2014 that community assemblages were altered. I have very few questions and comments, as I have seen one of the authors present a sub-set of this work previously and have discussed their methods etc in greater detail with them in person.

The data collected here to make these conclusions is probably second to none in terms of the temporal resolution and length compared to other existing time series in temperate kelp forest

ecosystems. This type of data set is what is required in many other locations and I applaud the authors for this long-term approach that would have taken thousands of man hours to complete. For example, in my home country there is perhaps only one data set of 3 or 4 years conducted in a similar way to this. The results are not overly novel, in that such impacts of marine heatwaves have been recorded previously, but the scientific community will gain insights from this work that would justify its publication in this outlet in my opinion.

In saying that, the manuscript is fairly short and it could have benefited from exploration of smaller details. For example, I am slightly biased here in that I was hoping to see greater exploration of trends within different seaweed groups. However, I understand the short format here and I consider the manuscript represents a useful description of the overall trends in community assemblages. The abstract should be altered to indicate the focus on the sessile invertebrates here. A greater focus on the presentation of giant kelp canopy cover could have also been interesting here for the reader.

We thank Reviewer #2 for their thoughtful assessment and feedback. We appreciate their interest in the dynamics of the understory macroalgal community and giant kelp canopy and agree that it is important! We agree that long term trends in community structure of understory macroalgae deserves further assessment, however, we feel that an analysis of these trends is outside the scope of this paper. Rather than expand our paper to include such analyses, we instead added a reference to Lamy et al. 2019 who describe changes in understory macroalgal species composition during the 2014-2016 warming event and Cavanaugh et al. 2019 who documented the response of the giant kelp canopy before and after the Blob. Per Reviewer 2's suggestion, we have altered the abstract to include sessile invertebrates in Line 8-13.

Specific Comments:

11. Line 161: 30 year periods are recommended in the Hobbday paper – but I can understand why in situ measurements at 18 years might be better than trying to rely on satellite derived estimates for longer time periods.

Yes, 30 years is the recommended length of time for establishing baseline temperature in a given region for defining heatwaves. However, as you correctly point out, we have used *in situ* measurements across the five sites for 18 years. Though satellite derived estimates can be used to estimate heatwave periods, temperature logger data used in this study took measurements at depth at each site every 15 minutes, thus providing more accurate estimates of average daily temperature at a finer spatial resolution.

12. Line 189: Spell out SEM first before using.

Thank you for catching this! It has now been included in text on Line 223.

13. Figure 3 is interesting, but what about the seaweeds?

See our response to general comments above. We plan to evaluate the response of understory algal community structure in a more in-depth analysis in a future publication!

Additional Changes:

We updated Figure 2 to include what was previously Supplemental Figure 1 as an additional panel:

Fig. 2. Piecewise structural equation modeling (SEM) for sessile invertebrate A) cover and B) species richness. Arrows indicate directionality of effects. Red arrows show negative relationships; black arrows show positive relationships. R² values are conditional R². Arrow widths are proportional to effect sizes as measured by standardized regression coefficients (shown next to arrows). *** = p<0.001, ** = p<0.01, * = p<0.05. Insignificant pathways are not included.

We replaced Supplemental Figure 1 to now include the mean \pm SE for the A) invertebrate percent cover and B) richness over time:

Supplementary Figure 1. Average seasonal (A) percent cover and (B) species richness of invertebrate phyla. Gray shading indicates anomalous warming period of spring 2014 to the winter of 2016. Missing data during the spring of 2020 corresponds to a data gap due to restrictions on research caused by the COVID-19 pandemic. Bars represent standard error.

REVIEWERS' COMMENTS:

Reviewer #2 (Remarks to the Author):

The authors have done a fantastic job addressing my comments/concerns on the previous version. The paper is very high in quality and importance. I recommend publication without further changes.